# Design, Synthesis and Structure—Activity Relationships of Phenylalanine-Containing Peptidomimetics as Novel HIV-1 Capsid Binders Based on Ugi Four-Component Reaction

**DOI:** 10.3390/molecules27185995

**Published:** 2022-09-14

**Authors:** Xiangkai Ji, Jing Li, Prem Prakash Sharma, Xiangyi Jiang, Brijesh Rathi, Zhen Gao, Lide Hu, Dongwei Kang, Erik De Clercq, Simon Cocklin, Chuanfeng Liu, Christophe Pannecouque, Alexej Dick, Xinyong Liu, Peng Zhan

**Affiliations:** 1Department of Medicinal Chemistry, Key Laboratory of Chemical Biology (Ministry of Education), School of Pharmaceutical Sciences, Cheeloo College of Medicine, Shandong University, 44 West Culture Road, Jinan 250012, China; 2Laboratory for Translational Chemistry and Drug Discovery, Department of Chemistry, Hansraj College, University of Delhi, Delhi 110007, India; 3China-Belgium Collaborative Research Center for Innovative Antiviral Drugs of Shandong Province, 44 West Culture Road, Jinan 250012, China; 4Laboratory of Virology and Chemotherapym, Rega Institute for Medical Research, K.U. Leuven, Herestraat 49 Postbus 1043 (09.A097), B-3000 Leuven, Belgium; 5Department of Biochemistry & Molecular Biology, Drexel University College of Medicine, Philadelphia, PA 19104, USA

**Keywords:** HIV-1, capsid modulators, peptidomimetics, Ugi four-component reaction, drug design

## Abstract

As a key structural protein, HIV capsid (CA) protein plays multiple roles in the HIV life cycle, and is considered a promising target for anti-HIV treatment. Based on the structural information of CA modulator **PF-74** bound to HIV-1 CA hexamer, 18 novel phenylalanine derivatives were synthesized via the Ugi four-component reaction. In vitro anti-HIV activity assays showed that most compounds exhibited low-micromolar-inhibitory potency against HIV. Among them, compound **I-19** exhibited the best anti-HIV-1 activity (EC_50_ = 2.53 ± 0.84 μM, CC_50_ = 107.61 ± 27.43 μM). In addition, **I-14** displayed excellent HIV-2 inhibitory activity (EC_50_ = 2.30 ± 0.11 μM, CC_50_ > 189.32 μM) with relatively low cytotoxicity, being more potent than that of the approved drug nevirapine (EC_50_ > 15.02 μM, CC_50_ > 15.2 μM). Additionally, surface plasmon resonance (SPR) binding assays demonstrated direct binding to the HIV CA protein. Moreover, molecular docking and molecular dynamics simulations provided additional information on the binding mode of **I-19** to HIV-1 CA. In summary, we further explored the structure—activity relationships (SARs) and selectivity of anti-HIV-1/HIV-2 of **PF-74** derivatives, which is conducive to discovering efficient anti-HIV drugs.

## 1. Introduction

Human immunodeficiency virus (HIV) is the causal agent of acquired immunodeficiency syndrome (AIDS) that poses a serious threat to global public health [1]. HIV consists of two genotypes, HIV-1 and HIV-2. HIV-1 is more virulent than HIV-2, and is mainly responsible for the major pandemic worldwide. However, the increasing risk of HIV-2 infection should also be taken seriously [2]. Combined antiretroviral therapy (cART) has been an effective method for anti-AIDS treatment in the past decades, transforming AIDS into a controllable chronic disease and successfully prolonging patients’ lives. However, long-term use of cART programs suffers from many risks, such as drug resistance, drug-drug interactions, severe toxicity, and other adverse reactions [3]. Therefore, there is still an urgent need to identify novel therapies for antiretroviral drugs and novel and unexplored mechanisms of action to achieve a cure for HIV infection. HIV capsid (CA) is an asymmetric fullerene-shaped cone comprised of about 1500 CA monomers. This cone-shaped lattice consists of approximately 250 hexamers and 12 pentamers [4]. CA monomer can be divided into the N-terminal domain (NTD) and C-terminal domain (CTD), which are connected by a flexible linker [5]. In the early stage of viral replication, the capsid proteins can interact with the host factors, such as nucleoporin 153 (NUP153), cleavage, and polyadenylation specificity factor-6 (CPSF6), to complete the uncoating process and help to complete the reverse transcription, nuclear import, and integration process [6]. In the late stage of viral replication, capsid proteins assemble into the recombinant capsid, encapsulating viral RNA and gag-pol precursor proteins. In addition, capsid proteins can also evade the immune system by interacting with numerous host factors [7,8]. Therefore, the capsid protein is a promising target for anti-HIV drug design and development due to its critical functions within the viral life cycle [9,10,11].

Numerous antivirals targeting HIV have been reported [12,13,14,15,16,17,18]. **PF-74** (**1**, Figure 1a), as the first small molecule cocrystallized with the HIV-1 CA protein, has received the most extensive attention and research [19]. During the early stage of HIV replication, **PF-74** can competitively, with host factors CPSF6 and NUP153, interact with HIV-1 CA, interfering with numerous viral processes, including viral uncoating, reverse transcription, nuclear import, and integration [7,8]. At a later stage, it adversely affects the stability of capsid and viral maturation by interfering with the normal late-stage CA assembly process [6]. However, its clinical application has been limited because of moderate activity (EC_50_ = 0.52–0.90 μM), a poor drug-like profile, and poor metabolic stability (T_1/2_ = 0.5–1.3 min) [17,20,21,22]. **GS-6207**, a **PF-74** derivative developed by Gilead Sciences and currently in phase III clinical trials, shows an increased affinity for CA by adding multiple hydrogen bond donors and acceptors. However, its clinical application is limited by its complex synthetic route, administration mode, and drug resistance [22,23,24].

Structural studies showed that the phenylalanine core of **PF-74** formed a wide range of hydrophobic interactions within the NTD region. The acylamino linker could form hydrogen bond interactions with Asn57 and Gln63. The nitrogen atom on the indole ring could form a hydrogen bond interaction with Gln63, and the indole ring could interact with Met66, Gln67, Lys70, Gln63 of the NTD, and Arg173 and Lys182 of the adjacent CTD [19,25,26]. Previous studies identified numerous novel **PF-74** derivatives with improved potency and metabolic stability [21,25,27,28,29,30]. Although they exhibited moderate anti-HIV activity (EC_50_ = 2.1–4.3 μM), they provided a basis for further modification of **PF-74** (Figure 1b). Our structure—activity relationship studies identified the phenylalanine core region of **PF-74** being crucial for maintaining antiviral activity. The indole group and linker portion acted on the solvent-exposed region with potential for synthetic modifications. Consequently, the linker portion of **PF-74** greatly influences anti-HIV activity.

A “Fraction of sp^3^ carbon atoms” (Fsp^3^, the number of sp^3^ hybridized carbons/total carbon count) has been proposed to measure the carbon saturation of molecules and to characterize the complexity of molecular space structures [31,32,33,34]. Researchers proposed and successfully proved that the increase in the saturation measured by Fsp^3^ within molecules improved the clinical success rate, which may be related to the increase in solubility or to the increased three-dimensional characteristics crucial to occupying the molecule’s binding site [35,36]. The multicomponent reaction refers to a reaction in which three or more reaction raw materials are used to synthesize a final product with each raw material fragment simultaneously. Compared with the traditional synthesis method, it can shorten the reaction process without separating the intermediates and effectively shorten the reaction time, improving the overall synthesis efficiency [37,38]. In addition, it plays an important role in constructing compound libraries. Therefore, using a fragment-based drug design strategy, we innovatively applied the Ugi four-component reaction to synthesize successfully for the first time, to our knowledge, peptidomimetic-based capsid protein binders with a linker rich in sp^3^ hybridization.

Herein, 18 novel **PF-74** derivatives were designed, synthesized, and evaluated for their anti-HIV potency. Furthermore, we demonstrated that the representative compound **I-19** interacts with CA using a surface plasmon resonance (SPR) binding assay. In addition, molecular dynamics simulations (MD) were performed on **I-19** to explore the binding mode with the CA hexamer. Moreover, we discuss our preliminary structure—activity relationships (SARs) of **PF-74** derivatives, which are essential for further improvement in antivirals targeting the HIV-1 CA protein.

## 2. Results and Discussion

### 2.1. Chemistry

As shown in Figure 1, Boc-*L*-3,5-difluorophenylalanine (**I-1**) amide was treated with *N*-methyl-4-aminoanisole via a condensation reaction to yield intermediate **I-2**. **I-2** was removed from the Boc group under the presence of trifluoroacetic acid to obtain intermediate **I-3**. Next, **I-3** was reacted with ammonium formate to give intermediate **I-4**, which was dehydrated to obtain the key intermediate isocyanide **I-5**. Finally, **I-5** was reacted with various acids, amines, and aldehydes to obtain the target compounds **I-6~I-23** via the Ugi four-component reaction [39].

### 2.2. Biological Activity

The newly synthesized **PF-74** derivatives were tested for anti-HIV activity and cytotoxicity in MT-4 cells using the MTT method. Meanwhile, **PF-74** and the approved drug nevirapine (NVP) were selected as controls. The evaluated results of EC_50_ (anti-HIV-1/2 activity), CC_50_ (cytotoxicity), and SI (selectivity index, CC_50_/EC_50_) are shown in Table 1.

The antiviral activity of this series of compounds against HIV-1 ranged from 30.29 to 2.53 μM. Among them, compound **I-19** (EC_50_ = 2.53 ± 0.84 μM) had the best antiviral activity against HIV-1, but was still weaker than **PF-74** (EC_50_ = 0.26 ± 0.08 μM). When R_1_ was *p*-fluorobenzyl, the activity of different substituents of R_2_ on HIV-1 was: 1-methyl-3-(trifluoromethyl)-tetrahydroindazole (**I-19**, EC_50_ = 2.53 ± 0.84 μM) ≈ 2,3-dimethyl-indole (**I-7**, EC_50_ = 2.79 ± 0.19 μM) ≈ 3-methyl-5-bromoindole (**I-9**, EC_50_ = 2.93 ± 0.32 μM) ≈ *p*-methylphenylboronic acid (**I-23**, EC_50_ = 4.33 ± 1.32 μM) > *p*-methylphenylboronic acid pinacol ester (**I-12**, EC_50_ = 11.73 ± 2.07 μM). Notably, when R_1_ was introduced into a smaller cyclopropyl group, the compounds (**I-11**, EC_50_ = 3.08 ± 0.15 μM; **I-18**, EC_50_ = 5.18 ± 1.11 μM) also possessed outstanding activity against HIV-1.

The activity of this series of compounds against HIV-2 ranged from 24.9 to 2.30 μM. Among them, **I-14** (EC_50_ = 2.30 ± 0.11 μM, CC_50_ > 189.32 μM) displayed the best anti-HIV-2 activity, which was equivalent to **PF-74** (EC_50_ = 2.22 ± 0.31 μM, CC_50_ = 73.83 ± 7.54 μM) and better than the approved drug nevirapine (EC_50_ > 15.02 μM, CC_50_ > 15.02 μM) and had lower toxicity than both. When R_1_ was *p*-cyanobenzyl, the activity of different substituents of R_2_ on HIV-2 was: 1-methylnaphthalene (**I-14**, EC_50_ = 2.30 ± 0.11 μM) > 3-methyl-5-fluoroindole (**I-15**, EC_50_ = 5.81 ± 1.11 μM) > 3-methyl-5-bromoindole (**I-16**, EC_50_ = 15.71 ± 13.30 μM) ≈ *p*-methylphenylboronic acid pinacol ester (**I-17**, EC_50_ = 16.16 ± 4.59 μM).

When R_1_ was *p*-fluorobenzyl and R_2_ was indole, the compounds (**I-7**, EC_50 (HIV-1)_ = 2.79 ± 0.19 μM, EC_50 (HIV-2)_ > 22.59 μM; **I-9**, EC_50(HIV-1)_ = 2.93 ± 0.32 μM, EC_50 (HIV-2)_ > 17.84 μM) had good selectivity to HIV-1. Furthermore, when R_2_ was 1-methylnaphthalene, the compounds (**I-15**, EC_50 (HIV-1)_ = 17.25 ± 9.98 μM, EC_50(HIV-2)_ = 5.81 ± 1.11 μM; **I-14**, EC_50(HIV-1)_ > 189.33 μM, EC_50 (HIV-2)_ = 2.30 ± 0.11 μM) had marked selectivity to HIV-2. It is worth noting that when the 1-methylnaphthalene of R_2_ is replaced by 2-methylnaphthalene, the anti-HIV activity of the compounds (**I-6**, **I-13**) will be lost.

Overall, the study of this series of compounds showed that the change in substituents greatly influenced antiviral activity and selectivity. In this series of compounds, the anti-HIV-2 activity of **I-14** was comparable to **PF-74** and better than the approved drug nevirapine, and the cytotoxicity was lower than that of both, which lays the foundation for the discovery of highly effective anti-HIV inhibitors.

### 2.3. Surface Plasmon Resonance (SPR) Assay on HIV-1 CA

In order to determine the target of the newly synthesized compounds, compound **I-19** was selected to verify the affinity with HIV-1 CA protein with an SPR experiment. **PF-74** was used as a positive control. As shown in Table 2, Figure 2 and Figure 3, the results indicated that the affinity of the compounds for CA proteins were **PF-74** (hexamer: *K*_D_ = 47.0 ± 0.6 nM; monomer: *K*_D_ = 968.5 ± 446.7 nM) > **I-19** (hexamer: *K*_D_ = 36866.7 ± 12012.6 nM; monomer: *K*_D_ = 13,338.0 ± 7793.4 nM), which is consistent with antiviral activity in vitro (**PF-74**, EC_50_ = 0.26 ± 0.08 µM > **I-19**, EC_50_ = 2.53 ± 0.84 µM). In contrast to **PF-74**, **I-19** is more inclined to interact with CA monomer. SPR experiments demonstrated that these compounds could be defined as specific HIV-1 CA modulators. The amino acid sequences **(GenBank accession number: M30895)** and NTD 3-D structure (PDB code: 2WLV) of HIV-2 CA protein are significantly similar to that of HIV-1 (GenBank accession number: AF324493; PDB code: 3H4E). In particular, the amino acids at the binding site of PF-74 are the same, except for amino acids 69 and 70. Therefore, these compounds could also bind to HIV-2 capsid protein.

### 2.4. Molecular Docking and MD Simulation

In order to explore the binding mode of these compounds with HIV-1 CA hexamer (PDB codes: 5TSX, 5HGL, https://www.rcsb.org), the representative compound **I-19** was selected for molecular docking; the MD simulation and the results were compared with PF-74. Compound I-19 exhibited a docking score, XP gscore, and binding free energy of −4.884 kcal/mol, −4.884 kcal/mol, and −37.23 kcal/mol, respectively. However, PF-74 showed a docking score, XP gscore, and binding free energy of −5.068 kcal/mol, −5.068 kcal/mol, and −57.72 kcal/mol, respectively. Compound I-19 showed slightly lower docking results in comparison to PF-74, which were in agreement with the experimental values where EC_50_ of I-19 and PF-74 was 2.53 ± 0.84 μM and 0.26 ± 0.08 μM, respectively. From the binding modes of **I-19** and **PF-74** with HIV-1 CA, the amide group of the phenylalanine backbone can form a hydrogen bond interaction with Asn57. The nitrogen atom on the indole substituent can form a hydrogen-bonding interaction with Gln63, while the nitrogen atom on the substituent group **I-19** can form a hydrogen-bonding interaction with Gln67 (Figure 4).

Figure 5 and 6 describe the stability of compounds **I-19** and **PF-74** in the binding pocket of HIV-1 CA (PDB code: 5TSX), respectively. In the **I-19**-5TSX complex, the protein structure became stable in the first few ns, and the average root mean square deviation (RMSD) of Cα was 3.8 Å (Figure 5a). For **I-19**, the structure of the capsid protein was stable at 35 ns, and the RMSD was 0.69 Å (Figure 5a). The root mean square fluctuation (RMSF) plot shows partial fluctuations in 4-fluorophenyl and indazole (Figure 5b). Compound **I-19** interacts with binding site residues through H bond (green), hydrophobic interaction (grey), and water bridge (blue). **I-19** forms an H bond with Asn57 and Lys70, hydrophobic interaction with Leu56, Lys70, and Ile73, and interaction with Asn53, Leu56, Asn57, and Lys70 through a water bridge (Figure 5c,d). Compound **I-19** maintained 6–7 binding forces with the binding site throughout the molecular dynamics simulation (Figure 5e).

In the **PF-74**-5TSX complex, the protein structure became stable at 40 ns with a RMSD of Cα of 5.7 Å. For **PF-74**, the structure of the capsid protein was stabilized at 42 ns with an RMSD of 3.4 ± 0.6 Å (Figure 6a). The RMSF plot of **PF-74** indicated that the indole moiety fluctuated the most during the simulation (Figure 6b). Compounds **I-19** and **PF-74** had similar binding when bound to proteins, but compound **PF-74** had a lower RMSD and RMSF than **I-19**. This indicates that compound **PF-74** has higher affinity and stability than **I-19**. Among them, **PF-74** can form hydrogen-bonding interactions with residues Asn53, Asn57, and Lys70 and can form hydrophobic interactions with residues Leu56, Met66, Lys70, and Ile73 (Figure 6c). The salt bridge interaction did not affect the stability of neither **I-19** nor **PF-74** molecules (Figure 6c). During the entire molecular dynamics simulation, **PF-74** maintained 6–7 binding forces with the binding site (Figure 6d).

The thermal-binding free energy of both the complexes was calculated (after ligand stability) for every 2 ns during simulation by MMGBSA. Compound **I-19** (−53.81 kcal/mol) showed better average binding energy compared to PF-74 (−44.45 kcal/mol) (Table 3). The H bond and water bridge formation (Figure 5c) during the simulation could play a role in compound **I-19′**s stability, which may be the reason for the better thermal-binding free energy of **I-19** over the PF-74 during the simulation.

Based on the molecular dynamics simulation, the conformations of CA-**I-19** and CA-**PF-74** complexes were verified utilizing a Ramachandran plot. No residues were present in the outlier region of the CA-**PF-74** and CA-**I-19** complex (Table 4).

In summary, MD studies further explored the binding mode of compounds **I-19** and **PF-74** with the HIV-1 CA hexamer (Figure 7). Importantly, these results are consistent with direct binding as judged by SPR and antiviral activity, evaluating our workflow and providing important guidance for further structural optimization efforts.

## 3. Conclusions

In this article, taking **PF-74** as the lead compound, 18 novel phenylalanine-containing peptidomimetics were designed and synthesized utilizing the Ugi four-component reaction. The results of antiviral activity in vitro showed that most compounds had single-digit micromolar antiviral activity against HIV-1. Four compounds, **I-7**, **I-9**, **I-19,** and **I-21,** displayed outstanding potency against HIV-1, with the EC_50_ values ranging from 2.93 μM to 2.53 μM, but there was still a certain gap compared with **PF-74**. The anti-HIV-2 activity of **I-14** (EC_50_ = 2.30 ± 0.11 μM, CC_50_ > 189.32 μM) was similar to that of **PF-74** (EC_50_ > 2.22 ± 0.31 μM, CC_50_= 73.83 ± 7.54 μM), and it was better than that of the marketed drug nevirapine (EC_50_ > 15.02 μM, CC_50_ > 15.02 μM); at the same time, it had lower toxicity than both, which deserves further research. Notably, when the hydrophobic naphthalene ring was introduced, the compounds had good selectivity for HIV-1/HIV-2. In addition, SPR-binding assays demonstrated that these compounds could be defined as HIV-1 CA modulators. The molecular dynamics simulation analysis indicated the potential binding mode between compound **I-19** and the NTD-CTD interface of the CA hexamer provided important guidance for further structural optimization. In summary, using a fragment-based drug design strategy, the Ugi four-component reaction was applied to the small molecule targeting the HIV capsid protein for the first time, which laid the foundation for the discovery of candidate drugs with excellent antiviral activity and optimized drug-like properties.

## 4. Materials and Methods

### 4.1. Chemistry

^1^H NMR and ^13^C NMR spectra were recorded on a Bruker Avance-400 NMR spectrometer (Standard G1313A instrument) with DMSO-*d*_6_ as the solvent. Chemical shifts were expressed in δ values (ppm) using tetramethylsilane (TMS) as an internal reference, and *J* values were expressed in hertz (Hz). Melting points of all the compounds were determined on a micromelting point apparatus and were uncorrected. Flash column chromatography was performed on a column packed with Silica Gel60 (200–300 mesh). Most of the solvents were obtained from Sinopharm Chemical Reagent Co, Ltd. (SCRC) and were AR grade. TLC was performed on silica Gel GF254 (Merck) and irradiated by ultraviolet light (*λ* = 254 nm). Flash column chromatography was performed on a column packed with Silica Gel60 (200–300 mesh).

#### 4.1.1. Tert-butyl (*S*)-(3-(3,5-difluorophenyl)-1-((4-methoxyphenyl) (methyl)amino)-1-oxopropan-2-yl) carbamate (**I-2**)

The starting material Boc-3,5-difluoro-*L*-phenylalanine mg, 6.64 mmol) and 4-methoxy-*N*-methylaniline mg, 6.64 mmol) were added to 20 mL dichloromethane and stirred in an ice bath for 30 min. Then, DIEA (2.23 mL, 13.29 mmol) and *N*-methyl-4-aminoanisole (610 mg, 4.43 mmol) were added and stirred at room temperature for another 8 h (monitored by TLC). The mixture was evaporated under reduced pressure, and the residue was washed with saturated sodium bicarbonate (50 mL) and extracted with ethyl acetate (3 × 10 mL). Then, the combined organic layer was washed with 1 N HCl, extracted with ethyl acetate (3 × 10 mL), dried with anhydrous Na_2_SO_4_, filtered, and concentrated under reduced pressure to afford corresponding crude intermediate **I-2** as a yellow oil with a yield of 60%. ^1^H NMR (400 MHz, DMSO-*d*_6_) δ 7.31 (d, *J* = 8.6 Hz, 2H, PhH), 7.05 (tt, *J* = 19.0, 9.0 Hz, 4H, PhH), 6.43 (d, *J* = 7.0 Hz, 2H, PhH), 4.14 (td, *J* = 10.6, 3.1 Hz, 1H, CH), 3.80 (s, 3H, OCH_3_), 3.15 (d, *J* = 7.8 Hz, 3H, NCH_3_), 2.87–2.73 (m, 1H, CH), 2.69–2.57 (m, 1H, CH), 1.29 (s, 9H, 3(CH_3_)). ESI-MS: *m*/*z* 443.78 [M + Na]^+^, C_22_H_26_F_2_N_2_O_4_ [420.4].

#### 4.1.2. (*S*)-2-amino-3-(3,5-difluorophenyl)-*N*-(4-methoxyphenyl)-*N*-methylpropanamide (**I-3**)

Intermediate **I-2** (500 mg, 1.19 mmol) was added to 30 mL dichloromethane, and then trifluoroacetic acid (410 mg, 3.57 mmol) was slowly dropped. The mixture was stirred at room temperature. After 1 h, the reaction was completed. The pH of the reaction solution was adjusted to 7 with saturated sodium bicarbonate solution, and 40 mL dichloromethane was added for extraction. The organic phase was separated, washed with saturated sodium chloride solution (3 × 20 mL), dried with anhydrous Na_2_SO_4_, filtered, and concentrated under reduced pressure to afford corresponding crude intermediate **I-3** as a yellow oil with a yield of 80%. ^1^H NMR (400 MHz, DMSO-*d*_6_) δ 7.07 (d, *J* = 8.3 Hz, 2H), 7.05–6.93 (m, 3H), 6.57 (h, *J* = 4.0 Hz, 2H), 3.78 (s, 3H), 3.35 (dd, *J* = 7.6, 5.9 Hz, 1H), 3.09 (s, 3H), 2.74 (dd, *J* = 13.1, 5.8 Hz, 1H), 2.54–2.45 (m, 2H), 1.82 (s, 2H).

#### 4.1.3. (*S*)-3-(3,5-difluorophenyl)-2-formamido-*N*-(4-methoxyphenyl)-*N*-methylpropanamide (**I-4**)

Intermediate **I-3** (340 mg, 1.06 mmol) was added to 10 mL acetonitrile, and then ammonium formate (130 mg, 2.12 mmol) was added to the solution, and heated at 90 °C for 24 h (monitored by TLC). After 24 h, the reaction was completed, filtered, and evaporated under reduced pressure. A total of 20 mL of water was added to the residue and extracted with ethyl acetate (3 × 10 mL). The organic phase was combined and washed with 20 mL saturated sodium chloride solution. The organic phase was dried with anhydrous Na_2_SO_4_, filtered, and concentrated under reduced pressure. The obtained crude product was separated by silica gel column chromatography to obtain 1.68 g pure yellow oil product of intermediate **I-4** with a yield of 60 %. ^1^H NMR (400 MHz, DMSO-*d*_6_) δ 8.47 (d, *J* = 8.3 Hz, 1H), 7.87 (s, 1H), 7.22 (d, *J* = 8.3 Hz, 2H), 7.05 (td, *J* = 8.9, 4.6 Hz, 4H), 6.50 (h, *J* = 4.3 Hz, 2H), 4.52 (td, *J* = 8.8, 4.5 Hz, 1H), 3.80 (s, 3H), 3.13 (s, 3H), 2.88 (dd, *J* = 13.6, 4.6 Hz, 1H), 2.67 (dd, *J* = 13.7, 9.4 Hz, 1H). ESI-MS: *m*/*z* 349.3 [M + 1]^+^, C_18_H_18_F_2_N_2_O_3_ [348.35]

#### 4.1.4. (*S*)-3-(3,5-difluorophenyl)-*N*-(4-methoxyphenyl)-*N*-methyl-2-(methylidyne-l4-azaneyl) propenamide (**I-5**)

The intermediate **I-4** (410 mg, 1.18 mmol) was added to 10 mL dichloromethane; then, triethylamine (360 mg, 3.54 mmol) was added to the solution, and phosphorus oxychloride (180 mg, 1.18 mmol) was slowly added under ice bath conditions, stirring at 0 °C for 24 h (monitored by TLC). After 24 h, the reaction was completed, quenched by ice water, extracted with dichloromethane (3 × 15 mL), combined with the organic phase, dried with anhydrous sodium sulfate, filtered, concentrated supernatant, mixed with silica gel, and subjected to column chromatography to obtain 170 mg pure product of light-yellow solid **I-5** with a yield of 45 %. ^1^H NMR (400 MHz, DMSO-*d*_6_) δ7.26–7.10 (m, 3H, PhH), 7.03 (d, *J* = 8.8 Hz, 2H, PhH), 6.76 (hept, *J* = 4.2 Hz, 2H, PhH), 4.48 (dd, *J* = 8.3, 5.8 Hz, 1H, CH), 3.80 (s, 3H, OCH_3_), 3.15 (s, 3H, NCH_3_), 3.14–3.07 (m, 1H, CH), 2.99 (dd, *J* = 13.6, 8.3 Hz, 1H, CH). ESI-MS: *m*/*z* 332.2 [M + 1]^+^, C_18_H_17_F_2_N_2_O_2_ [331.12].

#### 4.1.5. General Procedure for the Synthesis of Target Compounds **I-**(**6-23**)

Polyformaldehyde (18 mg, 0.20 mmol) was dissolved in 6.0 mL of anhydrous methanol, and different amines were dissolved in 6.0 mL of anhydrous methanol. Subsequently, different acids were added, stirred at room temperature for 10 min, and then the key intermediate I-**5** was added, heating reflux at 70 °C for 16 h (monitored by TLC). After the reaction was completed, 10 mL saturated sodium bicarbonate solution was added to the residue in the bottle, and 10 mL dichloromethane was used for extraction. The organic phase was separated, 10 mL 1N HCl solution was added for washing, and the organic phase was separated. After washing with 10 mL saturated sodium chloride solution, the organic phase was dried with anhydrous sodium sulfate, filtered, and concentrated under reduced pressure. The crude product was separated using silica gel column chromatography to obtain the target products **I-**(**6-23**).

##### (*S*)-3-(3,5-difluorophenyl)-2-(2-(*N*-(4-fluorobenzyl)-2-(naphthalen-2-yl)acetamido)acetamido)-*N*-(4-methoxyphenyl)-*N*-methylpropanamide (**I-6**):

Light-yellow solid, yield: 61%, mp: 175–177 ℃.^1^H NMR (400 MHz, DMSO-*d*_6_) δ 8.56 (d, *J* = 8.2 Hz, 1H), 7.93–7.86 (m, 1H), 7.86–7.80 (m, 2H), 7.69 (d, *J* = 46.7 Hz, 1H), 7.55–7.44 (m, 2H), 7.39–7.32 (m, 1H), 7.30–7.21 (m, 3H), 7.19–7.09 (m, 3H), 7.09–6.96 (m, 3H), 6.60–6.47 (m, 2H), 4.60 (td, *J* = 8.8, 7.9, 5.1 Hz, 1H), 4.53–4.20 (m, 2H), 3.96 (td, *J* = 27.6, 27.1, 13.4 Hz, 3H), 3.84–3.77 (m, 3H), 3.70 (s, 1H), 3.21–3.11 (m, 3H), 2.90 (ddd, *J* = 19.1, 14.3, 4.3 Hz, 1H), 2.75–2.62 (m, 1H). ^13^C NMR (100 MHz, DMSO-*d*_6_) δ 171.76, 170.90, 169.49 (d, *J* = 265.2 Hz), 162.51 (dd, *J* = 245.8, 13.4 Hz), 159.23, 159.15, 142.54 (t, *J* = 9.4 Hz), 135.84, 134.23 (d, *J* = 3.1 Hz), 133.87, 133.74, 133.46, 133.41, 132.25, 130.26 (d, *J* = 8.2 Hz), 129.48 (d, *J* = 8.1 Hz), 129.19, 128.49, 128.38, 127.98, 127.93, 127.89, 127.81, 126.48, 126.01, 115.84 (d, *J* = 21.4 Hz), 115.54 (d, *J* = 21.3 Hz), 115.33, 115.24, 112.41 (d, *J* = 24.4 Hz), 102.54, 55.94, 55.90, 51.47, 51.33, 49.85, 37.80, 37.34. ESI-MS: *m*/*z* 654.70 [M + 1]^+^, C_38_H_34_F_3_N_3_O_4_ [653.70].

##### (*S*)-3-(3,5-difluorophenyl)-2-(2-(*N*-(4-fluorobenzyl)-2-(2-methyl-1H-indol-3-yl)acetamido)acetamido)-*N*-(4-methoxyphenyl)-*N*-methylpropanamide (**I-7**):

light-yellow solid, yield: 10.20%, mp: 112.5-114.5℃. ^1^H NMR (400 MHz, DMSO-*d*_6_) δ 10.76 (d, *J* = 16.8 Hz, 1H), 8.51 (d, *J* = 8.1 Hz, 1H), 7.34 (d, *J* = 8.1 Hz, 1H), 7.25 (dd, *J* = 17.4, 8.2 Hz, 3H), 7.16 (dd, *J* = 8.2, 5.9 Hz, 2H), 7.07 (dd, *J* = 8.7, 5.2 Hz, 3H), 7.03–6.97 (m, 3H), 6.89 (t, *J* = 7.4 Hz, 1H), 6.50 (dd, *J* = 18.1, 6.7 Hz, 2H), 4.58 (tt, *J* = 11.3, 5.5 Hz, 1H), 4.51−4.40 (m, 1H), 4.18 (d, *J* = 15.1 Hz, 1H), 3.96 (d, *J* = 17.6 Hz, 1H), 3.87 (d, *J* = 12.3 Hz, 1H), 3.81 (s, 3H), 3.72 (d, *J* = 17.3 Hz, 1H), 3.56 (s, 1H), 3.15 (d, *J* = 11.0 Hz, 3H), 2.88 (ddd, *J* = 23.4, 13.6, 4.2 Hz, 1H), 2.74−2.60 (m, 1H), 2.22 (s, 3H). ^13^C NMR (150 MHz, DMSO-*d*_6_) δ 170.80, 170.21, 167.32, 161.49 (dd, *J* = 245.9, 13.1 Hz), 161.49 (dd, *J* = 254.9, 13.1 Hz), 159.20, 142.39, 135.22, 135.26, 134.26, 130.09 (d, *J* = 7.6 Hz), 129.74, 129.46 (d, *J* = 8.2 Hz), 129.18, 125.96, 125.90, 123.91, 123.82, 121.69, 121.59, 115.77, 115.66, 115.45, 115.38, 115.36, 115.20, 113.70, 113.69, 112.40, 112.42, 112.31, 112.28, 111.42, 108.50, 102.42, 102.28, 55.86, 51.50, 49.94, 49.18, 37.76, 37.31, 30.33, 15.32. ESI-MS: *m*/*z* 657.71 [M + 1]^+^, C_37_H_35_F_3_N_4_O_4_ [656.71].

##### (*S*)-3-(3,5-difluorophenyl)-2-(2-(*N*-(4-fluorobenzyl)-2-(naphthalen-1-yl)acetamido)acetamido)-*N*-(4-methoxyphenyl)-*N*-methylpropanamide (**I-8**):

light-yellow solid, yield: 10.20%, mp: 112.5–114.5 ℃. ^1^H NMR (400 MHz, DMSO-*d*_6_) δ 8.59 (d, *J* = 8.1 Hz, 1H), 7.92–7.79 (m, 3H), 7.52–7.41 (m, 3H), 7.26 (ddd, *J* = 14.5, 12.3, 7.2 Hz, 5H), 7.18–7.12 (m, 2H), 7.06 (d, *J* = 8.6 Hz, 2H), 6.94–6.83 (m, 1H), 6.53 (dt, *J* = 10.2, 5.0 Hz, 2H), 4.71–4.56 (m, 1H), 4.51 (t, *J* = 14.4 Hz, 1H), 4.21–4.11 (m, 2H), 4.05–3.93 (m, 3H), 3.80 (d, *J* = 7.6 Hz, 3H), 3.13 (s, 3H), 2.96–2.84 (m, 1H), 2.74–2.64 (m, 1H). ^13^C NMR (100 MHz, DMSO-*d*_6_) δ 171.75, 170.87, 168.54, 163.70 (d, *J* = 13.1 Hz), 163.06, 161.32, 159.23, 142.49, 135.87, 134.35, 133.75, 133.10 (d, *J* = 43.3 Hz), 130.25 (d, *J* = 8.1 Hz), 129.57 (d, *J* = 8.2 Hz), 129.21, 128.43 (d, *J* = 40.3 Hz), 127.59, 127.50, 126.36, 126.17, 126.04, 125.94, 125.89, 125.77, 125.09, 115.91 (d, *J* = 21.5 Hz), 115.54 (d, *J* = 21.2 Hz), 115.31, 112.49 (d, *J* = 6.4 Hz), 112.31 (d, *J* = 6.5 Hz), 102.49 (t, *J* = 25.7 Hz), 55.91, 51.67, 49.92, 49.27, 37.83, 37.45, 37.28. ESI-MS: *m*/*z* 654.5 [M + 1]^+^, C_38_H_34_F_3_N_3_O_4_ [653.70].

##### (*S*)-2-(2-(2-(5-bromo-1*H*-indol-3-yl)-*N*-(4-fluorobenzyl)acetamido)acetamido)-3-(3,5-difluorophenyl)-*N*-(4-methoxyphenyl)-*N*-methylpropanamide (**I-9**):

White solid, yield 9.60%, mp: 78–80 ℃. ^1^H NMR (400 MHz, DMSO-*d*_6_) δ 11.11 (d, *J* = 2.4 Hz, 1H), 8.57 (t, *J* = 8.3 Hz, 1H), 7.73–7.66 (m, 1H), 7.31 (dd, *J* = 13.5, 8.5 Hz, 4H), 7.23–7.17 (m, 4H), 7.11–7.06 (m, 3H), 6.56–6.47 (m, 3H), 5.47 (dd, *J* = 10.3, 7.2 Hz, 1H), 4.63–4.48 (m, 2H), 4.19 (dd, *J* = 15.0, 6.6 Hz, 1H), 4.05–3.91 (m, 2H), 3.80 (d, *J* = 1.5 Hz, 3H), 3.63 (d, *J* = 9.5 Hz, 2H), 3.15 (d, *J* = 9.5 Hz, 3H), 2.90 (td, *J* = 14.0, 6.6 Hz, 1H), 2.71 (dd, *J* = 13.5, 9.5 Hz, 1H). ^13^C NMR (150 MHz, DMSO-*d*_6_) δ 171.90, 170.86, 168.41, 162.54 (dd, *J* = 246.1, 13.2 Hz), 162.54 (dd, *J* = 246.1, 13.2 Hz), 159.23, 142.50, 135.86, 135.34, 134.32, 130.18 (d, *J* = 7.7 Hz), 129.75, 129.41 (d, *J* = 8.3 Hz), 129.16, 125.99, 125.91, 123.92, 123.86, 121.79, 121.64, 115.83, 115.69, 115.50, 115.36, 115.31, 115.26, 113.74, 113.71, 112.46, 112.43, 112.33, 112.30, 111.50, 108.51, 102.47, 102.30, 55.91, 51.55, 49.92, 49.15, 37.79, 37.34, 30.37. ESI-MS: *m*/*z* 757.26 [M + 2Na]^+^, C_36_H_32_BrF_3_N_4_O_4_ [721.58].

##### (*S*)-2-(2-(2-(5-bromo-1*H*-indol-3-yl)-*N*-(cyclohexylmethyl)acetamido)acetamido)-3-(3,5-difluorophenyl)-*N*-(4-methoxyphenyl)-*N*-methylpropanamide (**I-10**):

White solid, yield 9.80%, mp: 80–82 ℃. ^1^H NMR (400 MHz, DMSO-*d*_6_) δ 11.08 (d, *J* = 6.4 Hz, 1H), 8.51 (t, *J* = 9.1 Hz, 1H), 7.74–7.65 (m, 1H), 7.31 (dd, *J* = 8.5, 4.0 Hz, 2H), 7.26–7.13 (m, 2H), 7.11–6.90 (m, 3H), 6.52 (d, *J* = 7.0 Hz, 2H), 4.66–4.43 (m, 1H), 4.06–3.86 (m, 2H), 3.79 (d, *J* = 3.8 Hz, 3H), 3.77–3.67 (m, 1H), 3.54 (dd, *J* = 17.0, 7.8 Hz, 1H), 3.14 (d, *J* = 9.6 Hz, 3H), 3.12–3.03 (m, 1H), 2.95 (dd, *J* = 17.0, 5.3 Hz, 1H), 2.88 (dd, *J* = 17.0, 3.7 Hz, 1H), 2.75 (ddd, *J* = 26.8, 13.3, 8.3 Hz, 1H), 1.59–1.56 (m, 1H), 1.52–0.69 (m, 10H). ^13^C NMR (100 MHz, DMSO-*d*_6_) δ 171.70, 170.91, 168.75, 168.31, 162.50 (d, *J* = 232.9 Hz), 159.22, 135.86, 135.30 (d, *J* = 4.8 Hz), 129.67 (d, *J* = 2.3 Hz), 129.21, 125.82, 123.79, 121.85, 115.31, 112.56 (dd, *J* = 226.7, 7.3 Hz), 112.49, 112.25, 108.77, 102.48, 55.92, 52.81, 51.50, 51.02, 37.79, 37.24, 36.96, 36.04, 30.71, 30.48, 30.35, 26.52, 25.79. ESI-MS: *m*/*z* 726.6 [M + NH4^+^]^+^, C_36_H_39_BrF_2_N_4_O_4_ [709.63].

##### (*S*)-2-(2-(*N*-cyclopropyl-2-(2-methyl-1*H*-indol-3-yl)acetamido)acetamido)-*N*-(4-methoxyphenyl)-*N*-methyl-3-phenylpropanamide (**I-11**):

White solid, yield 9.80%, mp: 98–100 ℃. ^1^H NMR (400 MHz, DMSO-*d*_6_) δ 10.71 (s, 1H), 8.16 (d, *J* = 8.2 Hz, 1H), 7.34 (d, *J* = 7.8 Hz, 1H), 7.19 (dd, *J* = 8.2, 2.8 Hz, 3H), 6.99 (t, *J* = 8.1 Hz, 3H), 6.93 (t, *J* = 7.4 Hz, 1H), 6.85 (t, *J* = 7.2 Hz, 1H), 6.47 (d, *J* = 6.7 Hz, 2H), 4.45 (td, *J* = 8.8, 4.6 Hz, 1H), 3.94 (d, *J* = 16.5 Hz, 1H), 3.85 (d, *J* = 17.0 Hz, 2H), 3.76 (d, *J* = 5.4 Hz, 3H), 3.71 (s, 1H), 3.10 (d, *J* = 9.2 Hz, 3H), 2.83 (dd, *J* = 13.5, 4.1 Hz, 1H), 2.69–2.57 (m, 1H), 2.26 (s, 3H), 2.22–2.10 (m, 1H), 1.33–1.11 (m, 2H), 0.71–0.64 (m, 2H). ^13^C NMR (150 MHz, DMSO-*d*_6_) δ 171.82, 170.82, 168.36, 162.50 (dd, *J* = 246.1, 13.2 Hz), 159.21, 142.46, 135.81, 135.30, 130.16 (d, *J* = 7.7 Hz), 129.39 (d, *J* = 8.3 Hz), 125.96, 125.86, 123.91, 123.79, 121.76, 121.62, 115.82, 115.66, 115.48, 115.32, 113.70, 113.69, 112.45, 112.41, 112.30, 112.28, 111.49, 108.46, 102.45, 102.28, 55.90, 49.86, 49.12, 37.70, 37.30, 32.11, 30.32, 4.82, 4.82.ESI-MS: *m*/*z* 589.31 [M + 1]^+^, C_33_H_34_F_2_N_4_O_4_ [588.66].

##### (*S*)-3-(3,5-difluorophenyl)-2-(2-(*N*-(4-fluorobenzyl)-2-(4-(4,4,5,5-tetramethyl-1,3,2-dioxaborolan-2-yl)phenyl)acetamido)acetamido)-*N*-(4-methoxyphenyl)-*N*-methylpropanamide (**I-12**):

White solid, yield 10.5%, mp: 98–100 ℃. ^1^H NMR (400 MHz, DMSO-*d*_6_) δ 8.51 (d, *J* = 8.3 Hz, 1H), 7.59 (t, *J* = 6.8 Hz, 2H), 7.27–7.24 (m, 2H), 7.22–7.19 (m, 2H), 7.15 (dd, *J* = 10.9, 7.8 Hz, 4H), 7.06–7.00 (m, 3H), 6.50 (t, *J* = 6.7 Hz, 2H), 4.56 (ddd, *J* = 17.0, 10.9, 6.0 Hz, 1H), 4.42 (dd, *J* = 14.5, 7.6 Hz, 1H), 4.18 (d, *J* = 15.0 Hz, 1H), 3.96–3.88 (m, 2H), 3.82 (d, *J* = 8.3 Hz, 3H), 3.54 (s, 2H), 3.14 (d, *J* = 12.0 Hz, 3H), 2.94–2.81 (m, 1H), 2.73–2.61 (m, 1H), 1.29 (s, 12H). ^13^C NMR (150 MHz, DMSO-*d*_6_) δ 171.50, 170.80, 168.07, 162.68, 161.77, 161.68, 161.07, 159.24, 142.51, 142.45, 139.55, 139.44, 135.84, 134.88, 134.82, 134.77, 134.16 (d, *J* = 3.0 Hz), 130.24 (d, *J* = 8.1 Hz), 129.44 (d, *J* = 8.2 Hz), 129.33, 129.31, 129.24, 129.12, 115.86, 115.72, 115.50 (d, *J* = 21.3 Hz), 115.31, 115.25, 112.39 (d, *J* = 24.6 Hz), 102.45, 102.28, 84.00, 83.98, 55.89, 51.43, 49.79, 49.06, 47.69, 41.48, 37.76, 37.44, 25.38, 25.08. ESI-MS: *m*/*z* 730.16 [M + 1]^+^, 752.50 [M + Na]^+^, C_40_H_43_BF_3_N_3_O_6_ [729.60].

##### (*S*)-2-(2-(*N*-(4-cyanobenzyl)-2-(naphthalen-2-yl)acetamido)acetamido)-3-(3,5-difluorophenyl)-*N*-(4-methoxyphenyl)-*N*-methylpropanamide (**I-13**):

White solid, yield 7.80%, mp: 60–62 ℃. ^1^H NMR (400 MHz, DMSO-*d*_6_) δ 8.60 (d, *J* = 8.3 Hz, 1H), 7.86 (dt, *J* = 14.0, 6.9 Hz, 3H), 7.76 (t, *J* = 6.8 Hz, 2H), 7.67 (d, *J* = 22.8 Hz, 1H), 7.49 (p, *J* = 6.2, 5.6 Hz, 2H), 7.44–7.18 (m, 5H), 7.13–6.91 (m, 3H), 6.67–6.41 (m, 2H), 4.79–4.48 (m, 2H), 4.39 (d, *J* = 16.0 Hz, 1H), 4.20–3.95 (m, 2H), 3.82 (d, *J* = 19.2 Hz, 3H), 3.70 (dd, *J* = 16.1, 6.3 Hz, 2H), 3.15 (d, *J* = 14.7 Hz, 3H), 2.90 (td, *J* = 15.1, 14.5, 6.6 Hz, 1H), 2.70 (dd, *J* = 13.6, 10.0 Hz, 1H). ^13^C NMR (150 MHz, DMSO-*d*_6_) δ 172.08, 170.80, 168.16, 162.52 (dd, *J* = 246.0, 13.3 Hz), 159.25, 144.11, 142.49, 135.84, 133.75, 133.43, 132.86, 132.66, 132.30, 129.15, 128.88, 128.49, 128.27, 128.14, 128.07, 127.99, 127.97, 127.93, 127.90, 127.82, 126.48, 126.42, 126.02, 119.27, 115.34, 115.25, 112.40 (d, *J* = 24.9 Hz), 110.30, 102.50, 102.32, 55.94, 52.12, 51.49, 50.56, 50.17, 37.80, 37.41. ESI-MS: *m*/*z* 661.12 [M + 1]^+^, 683.52 [M + Na]^+^. C_39_H_34_F_2_N_4_O_4_ [660.72].

##### (*S*)-2-(2-(*N*-(4-cyanobenzyl)-2-(naphthalen-1-yl)acetamido)acetamido)-3-(3,5-difluorophenyl)-*N*-(4-methoxyphenyl)-*N*-methylpropanamide (**I-14**):

Light-yellow solid, yield 10.60%, mp: 106–108 ℃. ^1^H NMR (400 MHz, DMSO-*d*_6_) δ 8.63 (d, *J* = 8.1 Hz, 1H), 7.90 (d, *J* = 7.4 Hz, 1H), 7.87–7.76 (m, 4H), 7.53–7.45 (m, 2H), 7.41 (q, *J* = 10.3, 8.8 Hz, 3H), 7.34–7.20 (m, 3H), 7.04 (dd, *J* = 16.3, 8.8 Hz, 2H), 6.86 (d, *J* = 9.4 Hz, 1H), 6.51 (t, *J* = 6.4 Hz, 2H), 4.63–4.46 (m, 2H), 4.39–4.16 (m, 2H), 4.14–3.95 (m, 3H), 3.80 (d, *J* = 5.4 Hz, 3H), 3.12 (s, 3H), 2.89 (td, *J* = 14.9, 14.3, 6.7 Hz, 1H), 2.76–2.62 (m, 1H). ^13^C NMR (100 MHz, DMSO-*d*_6_) δ 172.85, 171.66, 169.36, 164.42, 162.04 (d, *J* = 13.0 Hz), 160.04, 145.11, 143.25 (t, *J* = 9.3 Hz), 135.61 (d, *J* = 212.5 Hz), 134.01, 133.74 (d, *J* = 12.6 Hz), 133.50, 130.02, 129.65, 129.28 (d, *J* = 32.1 Hz), 128.34, 126.99, 126.66 (d, *J* = 18.0 Hz), 125.90, 120.14, 116.12, 113.20 (d, *J* = 24.4 Hz), 111.03, 56.73, 52.51, 51.40, 51.15, 38.64, 38.17, 38.07. ESI-MS: *m*/*z* 683.54 [M + Na]^+^, C_39_H_34_F_2_N_4_O_4_ [660.72].

##### (*S*)-2-(2-(*N*-(4-cyanobenzyl)-2-(5-fluoro-2-methyl-1*H*-indol-3-yl)acetamido)acetamido)-3-(3,5-difluorophenyl)-*N*-(4-methoxyphenyl)-*N*-methylpropanamide (**I-15**):

Light-yellow solid, yield 10.50%, mp: 120–122 ℃. ^1^H NMR (400 MHz, DMSO-*d*_6_) δ 10.98 (d, *J* = 15.4 Hz, 1H), 8.57 (d, *J* = 8.1 Hz, 1H), 7.85–7.64 (m, 2H), 7.37–7.31 (m, 2H), 7.29–7.23 (m, 3H), 7.19 (dd, *J* = 14.2, 2.3 Hz, 1H), 7.09–6.96 (m, 3H), 6.91 (tq, *J* = 8.6, 2.6 Hz, 1H), 6.52 (d, *J* = 6.0 Hz, 2H), 4.67–4.45 (m, 2H), 4.42–4.02 (m, 2H), 4.02–3.91 (m, 1H), 3.80 (d, *J* = 2.8 Hz, 3H), 3.75 (s, 1H), 3.61 (s, 1H), 3.14 (d, *J* = 8.2 Hz, 3H), 2.89 (td, *J* = 15.5, 14.6, 6.6 Hz, 1H), 2.69 (ddd, *J* = 19.6, 11.9, 4.3 Hz, 1H). ^13^C NMR (100 MHz, DMSO-*d*_6_) δ 172.82, 171.51, 169.16, 164.42 (d, *J* = 13.0 Hz), 162.08 (d, *J* = 13.3 Hz), 159.94 (d, *J* = 10.1 Hz), 157.20 (d, *J* = 10.4 Hz), 143.35, 135.61, 136.25 (d, *J* = 8.1 Hz), 133.40, 130.08, 129.56, 126.75, 124.60, 122.48 (d, *J* = 21.0 Hz), 120.22, 116.03, 114.35, 113.08 (d, *J* = 24.6 Hz), 112.27, 110.89, 109.16, 103.31, 56.62, 52.34, 52.01, 51.36, 51.02, 38.58, 30.78. ESI-MS: *m*/*z* 668.34 [M + 1]^+^, C_36_H_32_F_4_N_4_O_4_ [667.24].

##### (*S*)-2-(2-(2-(5-bromo-2-methyl-1H-indol-3-yl)-*N*-(4-cyanobenzyl)acetamido)acetamido)-3-(3,5-difluorophenyl)-*N*-(4-methoxyphenyl)-*N*-methylpropanamide (**I-16**):

Light-yellow solid, yield 11.20%, mp: 120–122 ℃. ^1^H NMR (400 MHz, DMSO-*d*_6_) δ 11.08 (d, *J* = 13.8 Hz, 1H), 8.55 (d, *J* = 8.1 Hz, 1H), 7.73 (dd, *J* = 12.1, 8.2 Hz, 2H), 7.63 (d, *J* = 21.8 Hz, 1H), 7.35–7.16 (m, 6H), 7.09–6.91 (m, 3H), 6.51 (d, *J* = 5.9 Hz, 2H), 4.66–4.43 (m, 2H), 4.34 (d, *J* = 15.9 Hz, 1H), 4.19–3.88 (m, 2H), 3.84–3.73 (m, 4H), 3.62 (s, 1H), 3.14 (d, *J* = 10.0 Hz, 3H), 2.89 (td, *J* = 14.9, 14.2, 6.6 Hz, 1H), 2.67 (ddd, *J* = 17.7, 13.6, 9.5 Hz, 1H). ^13^C NMR (100 MHz, DMSO-*d*_6_) δ 172.96, 171.65, 169.23, 164.54 (d, *J* = 13.4 Hz), 162.09 (d, *J* = 13.3 Hz), 159.98 (d, *J* = 10.2 Hz), 145.11, 143.27, 136.63, 136.07 (d, *J* = 8.0 Hz), 133.40, 130.00, 129.60, 126.83, 124.67, 122.49 (d, *J* = 21.0 Hz), 120.11, 116.11, 114.53, 113.18 (d, *J* = 24.7 Hz), 112.28, 110.99, 109.20, 103.33, 56.74, 52.40, 52.11, 51.39, 51.04, 38.60, 30.99. ESI-MS: *m*/*z* 738.52 [M + 1]^+^, C_37_H_32_BrF_2_N_5_O_4_[737.20].

##### (*S*)-2-(2-(*N*-(4-cyanobenzyl)-2-(4-(4,4,5,5-tetramethyl-1,3,2-dioxaborolan-2-yl)phenyl)acetamido)acetamido)-3-(3,5-difluorophenyl)-*N*-(4-methoxyphenyl)-*N*-methylpropanamide (**I-17**):

Light-yellow solid, yield 11.20%, mp: 46–48 ℃. ^1^H NMR (400 MHz, DMSO-*d*_6_) δ 8.24 (d, *J* = 8.1 Hz, 1H), 7.60 (d, *J* = 7.6 Hz, 2H), 7.24 (t, *J* = 8.0 Hz, 4H), 7.02 (d, *J* = 8.4 Hz, 3H), 6.51 (d, *J* = 6.8 Hz, 2H), 4.47 (tt, *J* = 10.9, 5.4 Hz, 1H), 4.01–3.88 (m, 3H), 3.80 (s, 3H), 3.71 (d, *J* = 28.8 Hz, 1H), 3.13 (s, 3H), 2.87 (dd, *J* = 13.5, 4.2 Hz, 1H), 2.70–2.66 (m, 1H), 1.29 (s, 12H), 0.74–0.58 (m, 4H). ^13^C NMR (100 MHz, DMSO-*d*_6_) δ 170.76, 168.07, 162.49 (dd, *J* = 246.0, 13.4 Hz), 159.21, 144.08, 139.46, 135.79, 134.76, 132.89, 132.70, 129.41, 129.28, 129.16, 128.81, 128.10, 119.30, 115.31, 112.39 (d, *J* = 24.6 Hz), 110.24, 84.02, 74.00, 55.93, 55.91, 51.45, 50.41, 50.06, 37.78, 37.34, 25.42, 25.13. ESI-MS: *m*/*z* 759.56 [M + Na]^+^, C_41_H_43_BF_2_N_4_O_6_ [736.20].

##### (*S*)-2-(2-(*N*-cyclopropyl-2-(4-(4,4,5,5-tetramethyl-1,3,2-dioxaborolan-2-yl)phenyl)acetamido)acetamido)-3-(3,5-difluorophenyl)-*N*-(4-methoxyphenyl)-*N*-methylpropanamide (**I-18**):

Light-yellow solid, yield 11.20%, mp: 46–48 ℃. ^1^H NMR (400 MHz, DMSO-*d*_6_) δ 8.24 (d, *J* = 8.1 Hz, 1H), 7.60 (d, *J* = 7.6 Hz, 2H), 7.24 (t, *J* = 8.0 Hz, 4H), 7.02 (d, *J* = 8.4 Hz, 3H), 6.51 (d, *J* = 6.8 Hz, 2H), 4.47 (tt, *J* = 10.9, 5.4 Hz, 1H), 4.01–3.88 (m, 3H), 3.80 (s, 3H), 3.71 (d, *J* = 28.8 Hz, 1H), 3.13 (s, 3H), 2.87 (dd, *J* = 13.5, 4.2 Hz, 1H), 2.70–2.66 (m, 1H), 1.29 (s, 12H), 0.74–0.58 (m, 4H). ^13^C NMR (100 MHz, DMSO-*d*_6_) δ 173.20, 170.91, 168.57, 162.50 (dd, *J* = 245.7, 13.4 Hz), 159.14, 142.56, 139.85, 135.83, 134.70, 129.52, 129.15, 115.28, 112.50, 112.26, 102.43, 84.00, 55.91, 51.22, 49.51, 37.76, 37.39, 31.25, 25.42, 25.13, 9.03, 8.86. ESI-MS: *m*/*z* 662.17 [M + 1]^+^, 684.54 [M + Na]^+^, C_36_H_42_BF_2_N_3_O_6_ [661.31].

##### (*S*)-3-(3,5-difluorophenyl)-2-(2-(*N*-(4-fluorobenzyl)-2-(3-(trifluoromethyl)-4,5,6,7-tetrahydro-1*H*-indazol-1-yl)acetamido)acetamido)-*N*-(4-methoxyphenyl)-*N*-methylpropanamide (I-19):

Light-yellow solid, yield 11.20%, mp: 46–48 ℃. ^1^H NMR (400 MHz, DMSO-*d*_6_) δ 8.61 (d, *J* = 8.2 Hz, 1H), 7.28–7.21 (m, 4H), 7.15 (t, *J* = 8.8 Hz, 2H), 7.04 (d, *J* = 8.9 Hz, 2H), 7.01–6.95 (m, 1H), 6.51 (dd, *J* = 15.1, 7.4 Hz, 2H), 5.19 (s, 1H), 4.99 (s, 2H), 4.58–4.41 (m, 2H), 4.05 (d, *J* = 15.8 Hz, 2H), 3.97–3.86 (m, 1H), 3.79 (d, *J* = 7.0 Hz, 3H), 3.73–3.63 (m, 1H), 3.13 (d, *J* = 7.6 Hz, 3H), 2.95–2.83 (m, 1H), 2.67 (dd, *J* = 13.6, 9.6 Hz, 1H), 2.49–2.28 (m, 3H), 1.88–1.50 (m, 5H). ^13^C NMR (100 MHz, DMSO-*d*_6_) δ 170.70, 167.84, 167.62, 163.79, 161.35, 161.21, 159.21, 142.16, 135.78, 133.58, 130.33, 130.25, 129.82, 129.18, 115.89, 115.71, 115.49, 115.31, 114.50, 112.55, 112.48, 112.30, 102.51, 55.87, 51.61, 51.09, 49.41, 49.02, 37.82, 37.31, 22.41, 21.94, 20.90, 20.00. ESI-MS: *m*/*z* 716.28 [M + 1]^+^, C_36_H_35_F_6_N_5_O_4_. [715.26].

##### (*S*)-2-(2-(*N*-(cyclohexylmethyl)-2-(5-methyl-3-(trifluoromethyl)-1*H*-pyrazol-1-yl)acetamido)acetamido)-3-(3,5-difluorophenyl)-*N*-(4-methoxyphenyl)-*N*-methylpropanamide (**I-20**):

light-yellow solid, yield 11.20%, mp: 46–48 ℃. ^1^H NMR (400 MHz, DMSO-*d*_6_) δ 8.64 (d, *J* = 8.2 Hz, 1H), 7.26 (dd, *J* = 27.4, 8.7 Hz, 2H), 7.06–6.95 (m, 3H), 6.51 (dd, *J* = 17.3, 5.6 Hz, 3H), 5.23–4.92 (m, 2H), 4.61–4.47 (m, 1H), 4.13–3.91 (m, 2H), 3.79 (d, *J* = 6.2 Hz, 3H), 3.14 (d, *J* = 10.7 Hz, 5H), 2.91 (td, *J* = 15.9, 14.8, 6.8 Hz, 1H), 2.71 (dd, *J* = 13.6, 9.5 Hz, 1H), 2.14 (d, *J* = 20.2 Hz, 4H), 1.55–0.81 (m, 10H). ^13^C NMR (100 MHz, DMSO-*d*_6_) δ 170.76, 168.24, 167.24, 162.49 (dd, *J* = 245.8, 13.4 Hz), 159.22, 142.47, 135.80, 129.21, 129.13, 115.33, 115.25, 112.54, 112.30, 103.91, 102.50, 55.87, 53.12, 51.72, 51.57, 49.95, 37.82, 37.28, 35.98, 30.62, 30.52, 26.47, 25.77, 25.72, 10.90. ESI-MS: *m*/*z* 662.36 [M − 1]^-^, C_33_H_38_F_5_N_5_O_4_ [663.28].

##### (*S*)-2-(2-(*N*-benzyl-2-(2-methyl-1*H*-indol-3-yl)acetamido)acetamido)-3-(3,5-difluorophenyl)-*N*-(4-methoxyphenyl)-*N*-methylpropanamide (**I-21**):

Light-yellow solid, yield 11.50%, mp: 86–88 ℃. ^1^H NMR (400 MHz, DMSO-*d*_6_) δ 10.78 (d, *J* = 15.8 Hz, 1H), 8.53 (d, *J* = 8.1 Hz, 1H), 7.38 (d, *J* = 7.8 Hz, 1H), 7.29 (d, *J* = 8.7 Hz, 2H), 7.26–7.22 (m, 3H), 7.16–7.09 (m, 2H), 7.07 (d, *J* = 8.4 Hz, 2H), 7.04–6.97 (m, 3H), 6.94–6.88 (m, 1H), 6.59–6.44 (m, 2H), 4.62 (ddd, *J* = 16.5, 10.3, 5.7 Hz, 1H), 4.54–4.44 (m, 1H), 4.21 (d, *J* = 15.2 Hz, 1H), 4.04–3.89 (m, 1H), 3.87 (d, *J* = 9.5 Hz, 1H), 3.77 (d, *J* = 7.6 Hz, 1H), 3.59 (s, 1H), 3.16 (d, *J* = 11.9 Hz, 3H), 2.89 (ddd, *J* = 23.5, 13.7, 4.5 Hz, 1H), 2.68 (ddd, *J* = 28.5, 13.7, 9.6 Hz, 1H), 2.23 (d, *J* = 4.2 Hz, 3H). ^13^C NMR (150 MHz, DMSO-*d*_6_) δ 170.78, 170.20, 167.30, 161.46 (dd, *J* = 245.7, 13.0 Hz), 161.30, 159.21, 142.38, 135.20, 135.24, 134.24, 130.00 (d, *J* = 7.4 Hz), 129.72, 129.45 (d, *J* = 8.0 Hz), 129.16, 125.94, 125.91, 123.90, 123.80, 121.67, 121.56, 115.76, 115.64, 115.40, 115.32, 115.32, 115.13, 113.72, 113.64, 112.36, 112.43, 112.32, 112.24, 111.46, 108.56, 102.40, 102.26, 55.89, 51.54, 49.92, 49.16, 37.72, 37.32, 30.31, 15.32. ESI-MS: *m*/*z* 637.42 [M − 1]^-^, C_37_H_36_F_2_N_4_O_4_ [638.27].

##### (*S*)-2-(2-(*N*-benzyl-2-(naphthalen-2-yl)acetamido)acetamido)-3-(3,5-difluorophenyl)-*N*-(4-methoxyphenyl)-*N*-methylpropanamide (**I-22**):

Light-yellow solid, yield 10.50%, mp: 46–48 ℃. ^1^H NMR (600 MHz, DMSO-*d*_6_) δ 8.53 (d, *J* = 8.3 Hz, 1H), 7.89 (s, 1H), 7.83 (dd, *J* = 15.8, 8.8 Hz, 2H), 7.63 (s, 1H), 7.52–7.45 (m, 2H), 7.38–7.32 (m, 2H), 7.29 (dd, *J* = 8.0, 6.1 Hz, 3H), 7.26–7.23 (m, 1H), 7.15 (dd, *J* = 21.0, 7.2 Hz, 3H), 7.07–6.97 (m, 3H), 6.56–6.46 (m, 2H), 4.60 (td, *J* = 9.2, 8.1, 4.6 Hz, 1H), 4.34 (dd, *J* = 133.6, 15.0 Hz, 2H), 4.00–3.92 (m, 2H), 3.87 (d, *J* = 17.5 Hz, 1H), 3.80 (d, *J* = 8.4 Hz, 3H), 3.70 (s, 1H), 3.15 (d, *J* = 19.0 Hz, 3H), 2.89 (ddd, *J* = 27.5, 13.7, 4.3 Hz, 1H), 2.72–2.64 (m, 1H). ^13^C NMR (100 MHz, DMSO-*d*_6_) δ 171.70, 170.81, 168.18, 163.80, 163.67, 161.22, 159.22, 138.01, 135.85, 133.91, 133.42, 132.25, 129.21, 129.14, 128.84, 128.48, 128.40, 128.15, 127.94, 127.89, 127.80, 127.57, 127.41, 126.49, 126.01, 115.34, 115.25, 112.54, 112.30, 102.53, 55.94, 55.91, 51.49, 49.77, 49.56, 47.60, 37.82, 37.31. ESI-MS: *m*/*z* 637.38 [M + 1]^+^, C_38_H_35_F_2_N_3_O_4_[636.05].

##### (*S*)-(4-(2-((2-((3-(3,5-difluorophenyl)-1-((4-methoxyphenyl)(methyl)amino)-1-oxopropan-2-yl)amino)-2-oxoethyl)(4-fluorobenzyl)amino)-2-oxoethyl)phenyl) boronic acid (**I-23**):

White solid, yield 12.0%, mp: 123.6–125.8 ℃. ^1^H NMR (400 MHz, DMSO-*d*_6_) δ 8.52 (d, *J* = 8.2 Hz, 1H), 7.98 (s, 2H), 7.73 (d, *J* = 7.6 Hz, 2H), 7.27 (d, *J* = 8.5 Hz, 2H), 7.23–7.17 (m, 3H), 7.14–7.11 (m, 3H), 7.07–6.98 (m, 3H), 6.53 (d, *J* = 7.1 Hz, 2H), 4.63–4.50 (m, 1H), 4.42 (d, *J* = 14.8 Hz, 1H), 4.20 (d, *J* = 15.0 Hz, 1H), 3.92 (dd, *J* = 16.7, 6.9 Hz, 1H), 3.81 (s, 3H), 3.75 (d, *J* = 7.6 Hz, 1H), 3.52 (s, 2H), 3.15 (d, *J* = 11.2 Hz, 3H), 2.90 (dt, *J* = 17.7, 8.8 Hz, 1H), 2.75–2.61 (m, 1H). ^13^C NMR (150 MHz, DMSO-*d*_6_) δ 171.69, 170.79, 168.11, 163.35 (d, *J* = 13.3 Hz), 161.92 (dd, *J* = 243.3, 17.8 Hz), 161.72 (d, *J* = 13.3 Hz), 159.24, 142.53, 142.47, 137.90, 135.86, 134.56, 134.49, 134.22, 130.22 (d, *J* = 8.1 Hz), 129.50 (d, *J* = 8.2 Hz), 129.14, 128.75, 128.69, 115.81 (d, *J* = 21.3 Hz), 115.49 (d, *J* = 21.3 Hz), 115.34, 115.29, 112.39 (d, *J* = 24.4 Hz), 102.47, 102.30, 55.93, 51.42, 49.85, 49.05, 47.65, 37.79, 37.42. ESI-MS: *m*/*z* 648.52 [M + 1]^+^, C_34_H_33_BF_3_N_3_O_6_. [647.24].

### 4.2. In Vitro Anti-HIV Assay

The evaluation of the antiviral activity of the compounds against HIV in MT-4 cells was performed using the MTT assay as described below. Stock solutions (10 × final concentration) of test compounds were added in 25 μL volumes to two series of triplicate wells to allow simultaneous evaluation of their effects on mock- and HIV-infected cells at the beginning of each experiment. Serial 5-fold dilutions of test compounds were made directly in flat-bottomed 96-well microtiter trays using a Biomek 3000 robot (Beckman Instruments, Fullerton, CA). Untreated HIV- and mock-infected cell samples were included as controls. HIV stock (50 mL) at 100–300 CCID_50_ (50% cell culture infectious doses) or culture medium was added to either the infected or mock-infected wells of the microtiter tray. Mock-infected cells were used to evaluate the effects of the test compound on uninfected cells to assess the test compounds’ cytotoxicity. Exponentially growing MT-4 cells were centrifuged for 5 min at 220 g, and the supernatant was discarded. The MT-4 cells were resuspended at 6 × 10^5^ cells/mL, and 50 μL volumes were transferred to the microtiter tray wells. Five days after infection, the viability of mock- and HIV-infected cells was examined spectrophotometrically using the MTT assay. The MTT assay is based on the reduction of yellow-colored 3-(4,5-dimethylthiazol-2-yl)-2,5-diphenyltetrazolium bromide (MTT) (Acros Organics) by mitochondrial dehydrogenase activity in metabolically active cells to a blue-purple formazan that can be measured spectrophotometrically. The absorbances were read in an eight-channel computer-controlled photometer (Infinite M1000, Tecan) at two wavelengths (540 and 690 nm). All data were calculated using the median absorbance value of three wells. The 50% cytotoxic concentration (CC_50_) was defined as the concentration of the test compound that reduced the absorbance (OD540) of the mock-infected control sample by 50%. The concentration achieving 50% protection against the cytopathic effect of the virus in infected cells was defined as the 50% effective concentration (EC_50_).

### 4.3. Binding to CA Proteins Analysis via Surface Plasmon Resonance (SPR)

The CA hexamer was generated by introducing mutations at the following sites: A14C, E45C, W184A, and M185A through site-directed mutagenesis (Stratagene). While the A14C and E45C mutations stabilized the CA hexamer, the W184A and M185A prevented further oligomerization of hexamers into CA cones and tubes. All binding assays were performed on a ProteOn XPR36 SPR Protein Interaction Array System (Bio-Rad Laboratories, Hercules, CA, USA). The instrument temperature was set at 25 °C for all kinetic analyses. ProteOn GLH sensor chips were preconditioned with two short pulses each (10 s) of 50 mM NaOH, 100 mM HCl, and 0.5% sodium dodecyl sulfide. Then, the system was equilibrated with PBS-T buffer (20 mM sodium phosphate, 150 mM NaCl, and 0.005% polysorbate 20, pH 7.4). The surface of a GLH sensorchip was activated with a 1:100 dilution of a 1:1 mixture of 1-ethyl-3-(3-dimethylaminopropyl) carbodiimide hydrochloride (0.2 M) and sulfo-*N*-hydroxysuccinimide (0.05 M). Immediately after chip activation, the HIV-1 NL4-3 capsid protein constructs, purified as in the study by Xu et al. [40], were prepared at a concentration of 100 µg/mL in 10 mM sodium acetate, pH 5.0, and injected across ligand flow channels for 5 min at a flow rate of 30 µL/min. Then, after unreacted protein had been washed out, excess active ester groups on the sensor surface were capped by a 5 min injection of 1 M ethanolamine HCl (pH 8.0) at a flow rate of 5 µL/min. A reference surface was similarly created by immobilizing a nonspecific protein (IgG b12 anti HIV-1 gp120; obtained through the NIH AIDS Reagent Program, Division of AIDS, NIAID, NIH: Anti-HIV-1 gp120 Monoclonal (IgG1 b12) from Dr. Dennis Burton and Carlos Barbas) and was used as a background to correct nonspecific binding.

To prepare a compound for direct binding analysis, compound stock solutions, along with 100% DMSO, totaling 30 µL were made to a final volume of 1 mL by addition of sample preparation buffer (PBS, pH 7.4). Preparation of analyte in this manner ensured that the concentration of DMSO was matched with that of the running buffer with 3% DMSO. Serial dilutions were then prepared in the running buffer (PBS, 3% DMSO, 0.005% polysorbate 20, pH 7.4) and injected at a flow rate of 100 µL/min, for a 1 min association phase, followed by up to a 5 min dissociation phase using the “one shot kinetics” capability of the Proteon instrument [41]. Data were analyzed using the ProteOn Manager Software version 3.0 (Bio-Rad). The responses from the reference flow cell were subtracted to account for the nonspecific binding and injection artifacts. Experimental data were fitted to a simple 1:1 binding model (where applied). The average kinetic (association [k_a_] and dissociation [k_d_] rates) and equilibrium parameters generated from 3 replicates were used to define the on- and off-rates and equilibrium dissociation constant (K_D_). Please refer to the Appendix A for more details.

### 4.4. Molecular Docking and MD Simulation

The methodology followed in this study (molecular docking and MD simulation) was taken from our previous work [42]. There are reports available on structural similarities of NTD of HIV-1 CA (PDB code: 3H4E) and truncated HIV-2 CA NTD (PDB code: 2WLV). As “HIV-1 is more virulent than HIV-2” as well as the unavailability of the complete structure of HIV-2 CA protein, we selected HIV-1 CA as a target protein [43]. In order to further verify the accuracy of the docking observations, complexes of compound **I-19** and **PF-74** with capsid protein were selected for extensive 100 ns MD simulation. The structure of the gag-polyprotein (PDB: 5TSX, 5HGL) was obtained from the RCSB website (https://www.rcsb.org) for ligand-protein complex interaction analysis. The Schrödinger software (Desmond software, NY, USA) was used to perform the computational work. The structures were prepared prior to docking to remove structural errors. The compounds **I-19** and **PF-74** were prepared by the Ligprep tool prior to docking. Schrödinger suite inbuilt Epik module was used to predict the ionization states of all compounds at pH 7 ± 2 and tautomers generated. This in silico study was carried out under the OPLS2005 forcefield. Site-specific molecular docking of both compounds against HIV-1 capsid protein was performed at XP precision using the Glide module of Schrödinger suite. The binding site of capsid protein (PDB: 5TSX) was defined in reference to PDB: 5HGL. The Van der Waals radii scaling factor was 0.8, and the partial charge cutoff was set to 0.15. Both complexes were introduced into the Desmond software to study the binding stability of both compounds within their respective complex. These complexes were solvated in a TIP3P water model, and Na^+^ ions were added to neutralize both complexes. The stereo-chemical geometry of 5TSX protein residues was measured by a Ramachandran map by procheck. Please refer to the Appendix A for more details.

## Data Availability

Not applicable.

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
