# Peer review of "Design, Synthesis and Structure—Activity Relationships of Phenylalanine-Containing Peptidomimetics as Novel HIV-1 Capsid Binders Based on Ugi Four-Component Reaction"

_molecules, 2022, doi:10.3390/molecules27185995_

Round 1
Reviewer 1 Report
The present manuscript titled “Design, Synthesis and Structure-Activity Relationships of Phenylalanine-Containing Peptidomimetics as Novel HIV-1 Capsid Binders Based on Ugi Four-Component Reaction” is concise and presented well with the good interpretation of experimental results. Authors have utilized fragment-based drug design strategy and Ugi multi component reaction for the synthesis of novel phenylalanine-containing peptidomimetics small molecule targeting the HIV capsid protein for the first time. Four compounds I-7, I-9, I-19, and I-21 displayed outstanding potency against HIV-1, with the EC50 values ranging from 2.93 μM to 2.53 μM. Authors mentioned that the PF-74 has poor druglike profile and metabolic stability as hence doing pharmacokinetics study on the lead compounds is recommended for value addition to the current work. Therefore, this manuscript would be suitable to publish in Molecules journal after minor revisions as suggested
Minor comments:
1. Please correct the alignments of EC50 and CC50 values in the figure 1B.
2. Please provide the 1H and 13C NMR spectrum for all the final compounds.
3. Check all the references formatting: Some references title of the publication first letter is capitalize and, in some references, not.
Reviewer 2 Report
In this work, the authors have reported the design, synthesis and structure-activity relationships of Phenylalanine-Containing Peptidomimetics as Novel HIV-1 Cap-3 sid Binders. The authors further explored the structure-activity relationships and selectivity of anti-HIV-32 1/HIV-2 of PF-74 derivatives, which is conducive to the discovery of efficient anti-HIV drugs.
This manuscript is important and can be published. However, publication of this manuscript in its present form is not recommended. To be considered further for publication, this work will need further clarifications in support of the claims made in the paper. Some specific points of concern are noted below:
1) There are no methodological details of molecular dynamics simulation in this paper.
2) Modeled interactions are not fully characterized, quantitative information is missing, and the binding energy for the Phenylalanine-containing peptidomimetics could be easily extracted from MD simulation data.
Minor comments:
1) Figs. 5 and 6 needs to be improved. The residues along the X axis should be clear and readable.
Reviewer 3 Report
The manuscript by Ji et al describes a new approach to find anti-HIV compounds with the potential to find new therapies. The manuscript is well written and the work was well presented. However, I have some questions regarding this work that I would like to see addressed.
1 - The authors present results of their compounds to both HIV-1 and HIV-2, however, they don't explain whihc are the main differences between the 2 genotypes and if the CA- proteins are very different, which is of utmost importance to discuss the obtained results. For example, why I-14 works so well for HIV-2 but not for HIV-1.
2 - SPR experiments were only performed for I-19 and PF-74. Why? Other compounds (including NVP) should have been also tested to improve discussion. Also, it was not explained how it was possible to analyse monomer vs hexamer. I would expect that the monomers would oligomerize in solution or there is any condition that prevents oligomerization? If so, that was not well explained in the text. Also, it would be nice to have this experiment with HIV-2 CA.
3 - The MD simulation was performed only for I-19 and PF-74 with HIV-1 CA. The structure of HIV-2 CA is not available, is the same as HIV-1 CA (see comment 1)
4 - In the discussion, I-14 is referred as a good compound, however, no further studies were performed with it. Why?
Minor comments:
There is a supplementary file but it was not mentioned in the manuscript. Also, is does not have any legend, and it is confused. I supposed that it is the list with the chemical structure of all the compounds, which is missing is the main manuscript.
The figure legends should be improved in order to better understand the figures, namely Figs 1-4.
Round 2
Reviewer 3 Report
The revised manuscript by Ji et al is now improved and closer to be accepted for publication. All my major concerns were properly answered in the cover letter; however, I would like to see the text regarding questions 1 and 2 incorporated in the final manuscript. Also, the Sup File should be referred when the hexamer vs monomer part is introduced in the text, so the readers can easily find this information.
Author Response
Dear Reviewer,
The contents that need to be supplemented have been explained in sections 2.3,3.3,3.4 of the manuscript.
Best Wish.